# Colistin Resistance Genes in Broiler Chickens in Tunisia

**DOI:** 10.3390/ani13081409

**Published:** 2023-04-20

**Authors:** Antonietta Di Francesco, Daniela Salvatore, Sonia Sakhria, Fabrizio Bertelloni, Elena Catelli, Salma Ben Yahia, Aida Tlatli

**Affiliations:** 1Department of Veterinary Medical Sciences, University of Bologna, Ozzano dell’Emilia, 40064 Bologna, Italy; 2Institute of Veterinary Research of Tunisia, University of Tunis El Manar, Tunis 1006, Tunisia; 3Department of Veterinary Sciences, University of Pisa, 56124 Pisa, Italy

**Keywords:** antimicrobial resistance, colistin, *mcr* genes, chicken, PCR

## Abstract

**Simple Summary:**

The extensive use of colistin in livestock is recognized as the main cause of the emergence of colistin resistance in Gram-negative bacteria. This phenomenon represents a public health concern, as colistin is one of the last-resort antibiotics against multidrug-resistant deadly infections in human medicine. In the present survey, DNA extracted from cloacal swabs from 195 broiler chickens in Tunisia was tested by PCR for the ten mobilized colistin resistance (*mcr*) genes known so far. Of the 195 animals tested, 81 (41.5%) were *mcr*-1 positive. These results confirm the urgent nature of antimicrobial resistance in the Tunisian poultry sector and suggest the need for cautious use of colistin in the veterinary field.

**Abstract:**

Colistin is a polymyxin antibiotic that has been used in veterinary medicine for decades, as a treatment for enterobacterial digestive infections as well as a prophylactic treatment and growth promoter in livestock animals, leading to the emergence and spread of colistin-resistant Gram-negative bacteria and to a great public health concern, considering that colistin is one of the last-resort antibiotics against multidrug-resistant deadly infections in clinical practice. Previous studies performed on livestock animals in Tunisia using culture-dependent methods highlighted the presence of colistin-resistant Gram-negative bacteria. In the present survey, DNA extracted from cloacal swabs from 195 broiler chickens from six farms in Tunisia was tested via molecular methods for the ten mobilized colistin resistance (*mcr*) genes known so far. Of the 195 animals tested, 81 (41.5%) were *mcr*-1 positive. All the farms tested were positive, with a prevalence ranging from 13% to 93%. These results confirm the spread of colistin resistance in livestock animals in Tunisia and suggest that the investigation of antibiotic resistance genes by culture-independent methods could be a useful means of conducting epidemiological studies on the spread of antimicrobial resistance.

## 1. Introduction

Colistin (polymyxin E) is a polymyxin antibiotic commonly used in animal health, mainly orally, for the treatment of enterobacterial digestive infections in pigs, poultry and cattle [1]. In human medicine, since its first introduction in 1952 [2], colistin has mainly been used to treat Gram-negative infections, especially pseudomonal infections. In the early 1980s, the use of colistin was restricted in clinical settings in most parts of the world, because of the risk of neuro- and nephro-toxicity [3]. In recent years, the emergence of multidrug-resistant Gram-negative bacteria as well as the lack of development of new antimicrobial agents has led to the re-evaluation of the use of colistin in clinical practice as one of the last-resort antibiotics against multidrug-resistant deadly infections, in particular by strains resistant to carbapenems [4,5].

The widespread and excessive use of colistin has led to the emergence and spread of colistin resistance among Gram-negative bacteria. The livestock industry has been pointed out as the main source responsible for this phenomenon, considering the limited use of colistin in human medicine for two decades vs its extensive use in livestock production for the treatment of enterobacterial digestive infections as well as for prophylactic treatment and as a growth promoter [6,7]. Polymyxins, including colistin, are positively charged polypeptides that interact with the negatively charged phosphate group of lipid A of the lipopolysaccharide (LPS) of the outer membrane of Gram-negative bacteria, destabilizing the membrane to cause cell lysis [8]. One of the defence strategies most used by Gram-negative bacteria against polymyxins is the modification of the target LPS, which reduces its negative charge and attenuates its affinity for positively charged polymyxins [9]. Chromosomally encoded resistance mechanisms corresponding to mutations or deletions in genes involved in the biosynthesis of LPS have been known for some time [10]. Chromosomally encoded colistin resistance is of limited interest, as it is a rare, self-limiting mechanism, that is not transmissible within the population [11]. Of greater significance was the discovery in 2015 in China [12] of a mobilized colistin-resistance (*mcr*) gene, named *mcr*-1, which encodes an enzyme that catalyses the addition of phosphoethanolamine to the phosphate groups in lipid A, thus abolishing the negative charges to which cationic polymyxins have affinity. The emergence of *mcr*-1 has raised great concern, as it is located on a plasmid and is therefore potentially horizontally transferable between bacterial strains and species. Further, carriage of *mcr*-1 is often associated with co-carriage of other drug resistance genes, including those for carbapenemases and extended-spectrum β-lactamases [13]. To date, several *mcr* gene types, primarily *mcr*-1 and, less commonly, *mcr*-2 to *mcr*-10, and numerous variants [1], have been described worldwide in animals, food samples and humans [14]. All the *mcr* genes have been associated with conjugative plasmids, except *mcr*-6 which was found to be chromosomally located [15]. Among food-producing animals, a remarkable prevalence of colistin resistance has been highlighted in the poultry sector [16,17].

The aim of this study was to investigate, by PCR, the presence of *mcr* genes in broiler chickens in Tunisia.

## 2. Materials and Methods

### 2.1. Sampling

In this study we retrospectively examined DNA extracted from cloacal swabs collected in 2019 from 195 apparently healthy broiler chickens at two slaughterhouses in the governorate of Ben Arous (Grand Tunis, Tunisia). The chickens belonged to 13 lots from different poultry sheds on six farms (A–F), located in five governorates (Ben Arous, Bizerte, Béja, Zaghouan and Nabeul), inside a perimeter of 60 km. Each lot consisted of 15 randomly selected animals. All the farms were industrial, except for one rural chicken farm (Farm E/Lot 7).

Total DNA was extracted from each cloacal swab using the QIAamp DNA mini kit (Qiagen, Hilden, Germany) following the supplier’s recommendations. One extraction control, consisting of kit reagents only, was included.

### 2.2. Molecular Analysis

#### DNA Amplification and Sequencing

DNA samples were investigated by PCR targeting the genes *mcr*-1 to *mcr*-10. Each gene was amplified by an individual PCR, using primers described in Table 1.

The following PCR protocols were carried out: 5 min of initial denaturation at 94 °C followed by 35 cycles at 94 °C for 1 min, 51.3 °C (*mcr*-8 and *mcr*-9), 53 °C (*mcr*-1), 56 °C (*mcr*-7), 57 °C (*mcr*-2, *mcr*-3, *mcr*-4, *mcr*-6 and *mcr*-10), or 62 °C (*mcr*-5) for 1 min, and 72 °C for 1 min. A final extension step of 10 min at 72 °C completed the reaction. The DNA extracted from *Escherichia coli* field strains, containing colistin resistance plasmids, was used as a positive control. The extraction control and a distilled water negative control were also included.

The PCR products were analysed by 1% agarose gel electrophoresis; the DNA bands were stained with Midori Green Advance (Nippon Genetics Europe GmbH, Düren, Germany) and then visualized using ultraviolet (UV) trans illumination. The amplicons were purified using the High Pure PCR Product Purification Kit (Roche, Mannheim, Germany), and both DNA strands were sequenced (Bio-Fab Research, Rome, Italy). The sequences obtained were compared with the public sequences available using the BLAST server in GenBank database (National Center for Biotechnology Information 2023).

### 2.3. Statistical Analysis

The Fisher exact test and Chi-Square test were used to compare the positivity rate within and between farms (*p*-value < 0.05 was considered significant).

## 3. Results

The results are shown in Table 2.

Positive results were highlighted for the *mcr*-1 gene only (Figure 1).

Of the 195 animals tested, 81 (41.5%) were *mcr*-1 positive. All six farms showed positivity for *mcr*-1, with a different prevalence for each lot. The highest prevalence was highlighted on farm A (lot 1: 87%, 13/15; lot 2: 47%, 7/15), farm C (lot 4: 93%, 14/15; lot 6: 87%, 13/15) and farm F (lot 9: 73%, 11/15; lot 11: 87%, 13/15), in which both lots tested were positive. Farms B and D showed only one *mcr*-1 positive lot each, with a prevalence of 13% (lot 13: 2/15) and 33% (lot 10: 5/15), respectively. For farm E, the only lot sampled showed a prevalence of 20% (3/15). No statistical difference emerged among lots from the same farm (*p*-value > 0.05), whereas there was a significant difference between the *mcr*-1 positivity rate observed in farms A, C, and F and that highlighted in the other farms investigated (*p*-value < 0.05). The identity of the amplicons was confirmed by comparison between the sequences obtained and the corresponding sequences available in the GenBank database, showing 99–100% nucleotide similarity. A representative *mcr*-1 sequence was deposited in the GenBank database under accession number OQ439918.

## 4. Discussion

A global dispersion of *mcr* genes has been demonstrated, especially along the food chain [15,17], in both developed and developing countries, which are currently interconnected due to globalization of the trade of food animals and foodstuffs [22]. In recent years, the non-therapeutic use of colistin has been banned in several countries [23]. In Africa, except for South Africa, colistin is an over-the-counter medication sold and dispensed without a veterinarian’s supervision [22]. Based on the bibliographic data, colistin resistance genes have been detected in Northern (Tunisia, Algeria, Egypt, Morocco), Southern (South Africa), Central (Congo), Eastern (Tanzania, Sudan, Kenya) and Western (Nigeria) Africa, in humans, food animals and their products, wastewaters and terrestrial and aquatic wildlife, suggesting the dissemination of colistin resistance to all ecological niches [22]. Seven *mcr* variants—*mcr*-1, *mcr*-2, *mcr*-3, *mcr*-4, *mcr*-5, *mcr*-8 and *mcr*-9—were detected, with *mcr*-1 being dominant [22].

As far as North Africa is concerned, previous studies detected the highest prevalence of colistin resistance in Tunisia [7] where the presence of colistin-resistant Gram-negative bacteria has been well documented in clinical specimens [24], animals such as chickens [25], bovids [26], camels [27] and wild boars [28], retail meat [16], and wastewater [29]. Only the *mcr*-1 variant has been highlighted so far. The largest number of studies concerned the poultry industry where colistin has been widely used to prevent and treat *Enterobacteriaceae* infections [30]. Colistin resistance has been detected in *Escherichia coli* isolates from both apparently healthy chickens or poultry meat [16,25,31,32,33] and chickens that died due to colibacillosis [34], as well as in *Salmonella* spp. isolates in broiler flocks [35].

In this study we examined DNA extracted from cloacal swabs from 195 broiler chickens from six farms in Tunisia. Samples were analysed for the ten *mcr* genes known so far, highlighting *mcr*-1 positivity in 81 (41.5%) of 195 samples tested. All the farms sampled showed positivity for *mcr*-1, with prevalence ranging from 13% to 93%. A significantly higher *mcr*-1 prevalence rate was observed in farms A, C and F, when compared with the other farms tested, probably caused by increased or more frequent use of colistin and/or widespread circulation of selected colistin resistant bacterial strains. The results were concordant with the previous investigations cited above concerning the spread of colistin resistance in the Tunisian poultry sector, as well as detection of the *mcr*-1 gene, which is the only *mcr* gene type highlighted so far in all contexts examined in Tunisia.

In our opinion, this study shows two peculiarities, relating to the study approach and the number of *mcr* genes tested.

The phenomenon of antimicrobial resistance (AMR) is traditionally investigated using culture-dependent methods based on bacteriological culture and antibiotic susceptibility testing of isolated microorganisms. This approach is highly specific, but it could cause an underestimation of AMR occurrence due to a consistent non-culturable fraction of microorganisms or those that require a long period of growth. Recent studies introduced a molecular approach based on amplification of antimicrobial resistance target genes from environmental or biological samples [36,37,38,39,40,41,42,43]. This approach is more expensive than traditional cultivation and does not allow determination of the bacterial sources of resistance genes. However, it is a rapid method that avoids possible underestimation of the occurrence of AMR [44], as it is able to detect non-culturable or labile bacteria and provides more extensive information on the resistance patterns harboured by all bacteria present in the tested samples and not only those highlighted in selected colonies [45].

To our knowledge this study is the first investigation conducted in Tunisia in which all the *mcr* gene types known so far have been examined. In this regard, previous studies have shown a mismatch between phenotypic colistin resistance and the detection of *mcr* genes in some bacterial isolates [16,35]. The lack of evidence of *mcr* genes in DNA from colistin resistant isolates could be attributed to classic chromosomal mutations as well as the presence of untested *mcr* variants.

The samples examined in this study had previously been tested for 14 tetracycline resistance (*tet*) genes, showing a high frequency and diversity of *tet* genes [46]. The coexistence *of mcr* and *tet* genes is not surprising considering that both colistin and tetracycline have been widely used or abused in veterinary medicine [17].

Our results confirm the spread of colistin resistance in the Tunisian poultry sector. Given that the livestock industry has been pointed out as the main source responsible for the emergence and dissemination of colistin resistance, suggesting that animals may be an important source of transmission of colistin resistance to humans, continuous colistin resistance surveillance studies in the veterinary field, as well as a responsible use of the drug on farms, are needed, and farmers should be made aware of the potential dangers of self-medication. However, the reduction in the use of colistin must not lead to an increase in the use of other classes of critically important antimicrobial agents such as fluoroquinolones, third and fourth generation cephalosporins and macrolides, but must be achieved through the application of fundamental principles of good governance in animal health and good husbandry practices, e.g., taking care of animal welfare and applying strict biosecurity measures.

## 5. Conclusions

The emergence of colistin resistance from animal sources is a public health concern, as this antibiotic is considered to be the last line therapeutic option for infections caused by multidrug-resistant Gram-negative bacteria. As colistin is still widely used in veterinary medicine, especially in certain countries, continuous monitoring of mobile colistin resistance determinants in the veterinary field would be appropriate, to trace the dissemination of *mcr* genes and provide a more precise assessment of the risk of food-borne antimicrobial resistance [23].

## Figures and Tables

**Figure 1 animals-13-01409-f001:**
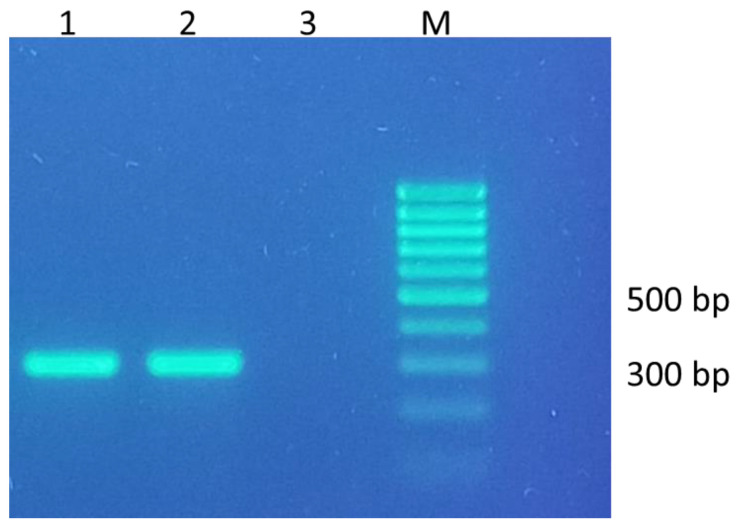
PCR amplicons. Lane 1, 308 bp *mcr*-1 gene fragment; lane 2, positive control; lane 3, distilled water negative control; lane M, MassRuler Low Range DNA Ladder (Thermo Fisher Scientific, Vilnius, Lithuania).

**Table 1 animals-13-01409-t001:** PCR primers used in this study.

Primer Name	PCR Primer Sequence 5′-3′	Amplicon Size (bp)	Target Gene	Reference
MCR1-F MCR1-R	5′-CGGTCAGTCCGTTTGTTC-3′ 5′-CTTGGTCGGTCTGTAGGG-3′	308	*mcr*-1	[12]
MCR2-F MCR2-R	5′-CTGTTGCTTGTGCCGATTGGACTA-3′ 5′-ACGGCCATAGCCATTGAACTGC-3′	282	*mcr*-2	[18]
MCR3-F MCR3-R	5′-CGCTTATGTTCTTTTTGGCACTGTATT -3′ 5′-TGAGCAATTTCACTATCGAGGTCTTG-3′	1063	*mcr*-3	[18]
MCR4-F MCR4-R	5′-AATTGTCGTGGGAAAAGCCGC-3′ 5′-CTGCTGACTGGGCTATTACCG-3′	1062	*mcr*-4	[18]
MCR5-F MCR5-R	5′-GTGAAACAGGTGATCGTGACTTACCG-3′ 5′-CGTGCTTTACACCGATCATGTGCT-3′	271	*mcr*-5	[18]
MCR6-F MCR6-R	5′-GTCCGGTCAATCCCTATCTGT-3′ 5′-ATCACGGGATTGACATAGCTAC-3′	556	*mcr*-6	[19]
MCR7-F MCR7-R	5′-TGCTCAAGCCCTTCTTTTCGT-3′ 5′-TTCATCTGCGCCACCTCGT-3′	892	*mcr*-7	[19]
MCR8-F MCR8-R	5′-TTGTCGTCGTGGGCGAAAC-3′ 5′-CTGTCGCAAGTTGGGCTAAAG3′	514	*mcr*-8	[20]
MCR9-F MCR9-R	5′-CGGCGAACTACGCTTACAG-3′ 5′-CGCACAGTTTCGGGTTATCAC-3′	465	*mcr*-9	[20]
MCR10-F MCR10-R	5′-GGACCGACCTATTACCAGCG-3′ 5′-GGCATTATGCTGCAGACACG-3′	365	*mcr*-10	[21]

**Table 2 animals-13-01409-t002:** Number of cloacal swabs PCR positive for colistin resistance genes *mcr*-1 to *mcr*-10. (Each lot consisted of 15 animals).

Farm/Lot	*mcr*-1	*mcr*-2	*mcr*-3	*mcr*-4	*mcr*-5	*mcr*-6	*mcr*-7	*mcr*-8	*mcr*-9	*mcr*-10
A/1	13	0	0	0	0	0	0	0	0	0
A/2	7	0	0	0	0	0	0	0	0	0
B/3	0	0	0	0	0	0	0	0	0	0
B/13	2	0	0	0	0	0	0	0	0	0
C/4	14	0	0	0	0	0	0	0	0	0
C/6	13	0	0	0	0	0	0	0	0	0
D/5	0	0	0	0	0	0	0	0	0	0
D/8	0	0	0	0	0	0	0	0	0	0
D/10	5	0	0	0	0	0	0	0	0	0
D/12	0	0	0	0	0	0	0	0	0	0
E/7	3	0	0	0	0	0	0	0	0	0
F/9	11	0	0	0	0	0	0	0	0	0
F/11	13	0	0	0	0	0	0	0	0	0
Total	81	0	0	0	0	0	0	0	0	0
N (%)	(41.5%)	0%	0%	0%	0%	0%	0%	0%	0%	0%

## Data Availability

The sequence generated in this study is available in GenBank under Accession number OQ439918.

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
