# Peer review of "Colistin Resistance Genes in Broiler Chickens in Tunisia"

_animals, 2023, doi:10.3390/ani13081409_

Round 1

Reviewer 1 Report

1.- In material and methods it  lacks the statistical method or statistical evaluation.

2.- In table 2 there is an error in the digits

3.- Due to the shallowness of the data, it would be necessary to add an image of amplified product as a control.

4.- it remains to add sensi tivity and specificity of your test

Author Response

Dear Reviewer,

thank you for your suggestions which will certainly improve the quality of the paper.

Taking your comments into account, the text has been modified as follows:

1.- In material and methods it lacks the statistical method or statistical evaluation.

A statistical evaluation has been added.

2.- In table 2 there is an error in the digits

Table 2 has been modified.

3.- Due to the shallowness of the data, it would be necessary to add an image of amplified product as a control.

An image of the mcr-1 amplicon has been added.

4.- it remains to add sensitivity and specificity of your test

We have not included an evaluation of the sensitivity and specificity of the PCRs performed since we have used already published primers and protocols whose references are shown in Table 1. In addition, the specificity of the PCR used to amplify the mcr-1 gene was confirmed by the sequencing results of the amplicons, which showed 99-100% nucleotide similarity with public mcr-1 sequences available in GenBank database.

Reviewer 2 Report

This paper simply indicated the distribution of the mcr gene in feces.

The importance of the mcr gene is huge. However, there is no point in only showing the presence or absence of genes without antibiotic sensitivity tests or any other experiments.

This paper is meaningless without additional experiments.

Author Response

Dear Reviewer,

having read your opinion so tranchant on the quality of the method we used and considering that our observations on the limits and merits of the culture-independent approach (lines 176-187), evidently have not been considered exhaustive, I enclose below the observations of other Authors who have taken into consideration the molecular approach:

Farooq et al. Antibiotic resistance genes occurrence in conventional and antibiotic-free poultry farming, Italy. Animals (Basel) 2022, 12, 2310. “More recently, a culture-independent approach has been applied to investigate the ARGs distribution in chicken intestinal microbiota, performing PCR-based or metagenomic sequence analysis [11 – 13 ]. These alternative methods appear to be particularly useful to study the microbial community and its genetic profiles in different types of samples, such as animal manure or agricultural soils, providing more extensive information on the resistance patterns harboured by all bacteria from investigated samples and not only in selected colonies.”

Galhano et al. Antimicrobial resistance gene detection methods for bacteria in animal-based foods: a brief review of highlights and advantages. Microorganisms 2021, 9, 923. “In this context, there are several methods of identifying resistant bacteria (phenotypic resistance) and their respective resistance genes (genotypic resistance). The traditional methods are primarily based on the culture of these microorganisms under specific conditions. Although simple and easy to carry out, some aspects are not so advantageous. The existence of viable non-cultivable microorganisms, or the long time that certain microorganisms may have to multiply in the environment, ends up being an obstacle to some researches [ 14,25 –27 ]. On the other hand, molecular methods are essentially based on the amplification of target genes (i.e., PCR, real-time PCR (qPCR), multiplex PCR, random amplified polymorphic DNA (RAPD), PCR combined with restriction fragment length polymorphism (PCR–RFLP)), whole-genome sequencing and metagenomics [28 –30]. Although more expensive than traditional cultivation, they are essential tools for the study of multiple microbiomes.”

Blanco-Peña et al. Antimicrobial resistance genes in pigeons from public parks in Costa Rica. Zoonoses Public Health 2017, 64, e23–e30. Generally used diagnostic methods that involve bacterial culture and subsequent detection of ARGs in the isolates could affect the results of analysis because non-culturable fractions include most of the microbial community (only 0.1–1% of soil bacteria can be grown in laboratory conditions) (Torsvik and Øvreas, 2002). This hinders the under-standing of potential negative effects of AMR on the ecosystem and on human and animal health, and is the reason for which some authors recommend genetic studies rather than those based only on cultures (Jiang et al., 2013; Marti et al., 2013).”

Marti et al.2013: Prevalence of antibiotic resistance genes and bacterial community composition in a river influenced by a wastewater treatment plant. PLoS ONE8, e78906. “Although antibiotic resistance studies have been focused on cultivable bacteria and/or indicator organisms in treated wastewater, the vast majority of environmental bacteria cannot be cultured under standard laboratory condition.………………………………. We used culture-independent approaches to determine the prevalence of ARGs and to examine how bacterial communities from biofilms and sediments respond to the discharge of WWTP effluents in the receiving river.”

Di Francesco et al. Research Note: Detection of antibiotic-resistance genes in commercial poultry and turkey flocks from Italy. Poultry Science. 2021 May;100(5):101084. “Current methods applied for monitoring of AMR are mainly based on culturing indicator bacteria followed by phenotypic AMR determination. This procedure targets a limited number of species and isolates present in the microbiota and, therefore, probably represents only a fraction of its resistome (the collective pool of ARG). On the contrary, the biomolecular approaches used in recent studies may represent an alternative tool to monitoring and verifying the presence of ARG in environments with high microbial density such as intensive farming.”

Smoglica et al. Occurrence of the tetracycline resistance gene tetA(P) in Apennine wolves (Canis lupus italicus) from different human–wildlife interfaces. J. Global Antimicrob. Res. 2020, 23, 184−185. “In order to identify the genetic determinants of antimicrobial resistance, the application of culture-independent approaches (PCR or metagenomics assay) are powerful and quick tools to investigate the ARGs occurring in wildlife.”

Vittecoq et al. Antimicrobial resistance in wildlife. J. Appl. Ecol. 2016;53:519–529. doi: 10.1111/1365-2664.12596. “Considering AMR genes as environmental contaminants and using methods that allow searching directly for these genes rather than for the bacteria carrying them may help in efficiently following the spread of AMR in all the components of the AMR transmission network. For example, methods involving polymerase chain reaction (PCR) use could permit searching for specific AMR genes whose spread is particularly worrying or that could be used as general AMR contamination markers (Prudenet al.2006; Gillingset al.2015).”

Singer et al. Can landscape ecology untangle the complexity of antibiotic resistance? Nat. Rev. Microbiol. 2007;4:943–952. “Consequently, analyses of antibiotic-resistance emergence, dissemination and persistence might be better conducted at the level of the gene.”

Zhu et al. Diverse and abundant antibiotic resistance genes in Chinese swine farms. Proc. Natl. Acad. Sci. USA. 2013;110:3435–3440. “Although antibiotic-resistant bacteria have been isolated and characterized from farm soils (21, 25), this method only samples microbes that are culturable and express their ARGs under those conditions. ARGs of noncultured populations, as well as “silent” or unexpressed ARGs (26), are sources of risk because they contribute to the resistance reservoir and could be horizontally transferred or expressed under other conditions.”

I would add that the suggestion of carrying out "additional experiments" on cloacal swabs taken 4 years ago for molecular purposes (therefore not stored during transport and stay at customs in environmental conditions suitable for bacterial cultures) seems to me unlikely to be implemented.

For your information, the paper had undergone professional linguistic proofreading by Proof-reading-service.com before being sent to Animals.
